# Hot Deformation Behavior and Microstructure Evolution of Fe–5Mn–3Al–0.1C High-Strength Lightweight Steel for Automobiles

**DOI:** 10.3390/ma14102478

**Published:** 2021-05-11

**Authors:** Guangming Liu, Jinbin Wang, Yafeng Ji, Runyuan Hao, Huaying Li, Yugui Li, Zhengyi Jiang

**Affiliations:** 1School of Materials Science and Engineering, Taiyuan University of Science and Technology, Taiyuan 030024, China; liugm@tyust.edu.cn (G.L.); kdzghry@vip.163.com (R.H.); lihuay@tyust.edu.cn (H.L.); liyugui2008@163.com (Y.L.); 2Coordinative Innovation Centre of Taiyuan Heavy Machinery Equipment, Taiyuan University of Science and Technology, Taiyuan 030024, China; 3State Key Laboratory of Rolling and Automation, Northeastern University, Shenyang 110819, China; wjb0226@sina.com; 4School of Mechanical Engineering, Taiyuan University of Science and Technology, Taiyuan 030024, China; 5School of Mechanical, Materials, Mechatronic and Biomedical Engineering, University of Wollongong, Wollongong, NSW 2522, Australia; jiang@uow.edu.au

**Keywords:** hot deformation behavior, microstructure evolution, automobile steel, nucleation mechanism, dynamic recrystallization

## Abstract

The hot deformation behavior of a newly designed Fe–5Mn–3Al–0.1C (wt.%) medium manganese steel was investigated using hot compression tests in the temperature range of 900 to 1150 °C, at constant strain rates of 0.1, 1, 2.5, 5, 10, and 20 s^−1^. A detailed analysis of the hot deformation parameters, focusing on the flow behavior, hot processing map, dynamic recrystallization (DRX) critical stress, and nucleation mechanism, was undertaken to understand the hot rolling process of the newly designed steel. The flow behavior is sensitive to deformation parameters, and the Zener–Hollomon parameter was coupled with the temperature and strain rate. Three-dimensional processing maps were developed considering the effect of strain and were used to determine safe and unsafe deformation conditions in association with the microstructural evolution. In the deformation condition, the microstructure of the steel consisted of δ-ferrite and austenite; in addition, there was a formation of DRX grains within the δ-ferrite grains and austenite grains during the hot compression test. The microstructure evolution and two types of DRX nucleation mechanisms were identified; it was observed that discontinuous dynamic recrystallization (DDRX) is the primary nucleation mechanism of austenite, while continuous dynamic recrystallization (CDRX) is the primary nucleation mechanism of δ-ferrite. The steel possesses unfavorable toughness at the deformation temperature of 900 °C, which is mainly due to the presence of coarse κ-carbides along grain boundaries, as well as the lower strengthening effect of grain boundaries. This study identified a relatively ideal hot processing region for the steel. Further exploration of hot roll tests will follow in the future.

## 1. Introduction

Fe–Mn–Al–C type advanced high strength steels (AHSSs) have attracted considerable attention in the automobile industry due to their low density, excellent mechanical properties, and desirable fatigue properties, which could contribute to improvements in fuel economy and a decrease of exhaust emissions [1,2]. The main challenge faced by automotive steels is to find a balance between safety, the environment, energy, and cost. Previous studies have suggested that the use of AHSSs instead of conventional steels in a 1560 kg family car can save 390 kg of weight, and can reduce cost and pollutant discharge by 14% and 5.7%, respectively [3,4]. In conventional AHSSs, Si could be partially replaced by Al, since Al plays a similar role in suppressing the formation of cementite. Moreover, the addition of Al can be an effective way to decrease the densities of steels, which makes Al-bearing AHSSs a very promising low-density steel. Three generations of AHSSs have already been developed. The first generation AHSSs have a tensile strength higher than 600 MPa, however, their low elongation (<25%) limits their applications in the automobile industry [5]. Second generation AHSSs have a higher tensile strength of 800–1700 MPa and their total elongation can reach 35–60%, however, the addition of a high content of alloying elements increases the cost, and may cause difficulties in welding [6]. The third generation AHSSs demonstrate intermediate mechanical properties and their tensile strength and elongation could be up to 0.4–1.0 GPa and 30–100%, respectively. Third generation AHSS are usually lower cost and have a lower density compared to the previous two generations.

To date, only limited third generation AHSS are being produced by iron and steel companies such as Baosteel, JFE Steel, and POSCO, owing to the difficulties in microstructural control and processing techniques [4,5,6,7]. Hot deformation is known as an effective way to investigate hot roll behaviors and microstructure evolution. Grajcar et al. [8] investigated the hot-working behavior and microstructure properties relationships of four experimental steels with various Mn and Nb contents by continuous compression, double–hit compression, and seven–step compression tests. It was found that the increase in Mn content was helpful to improve the softening kinetics of austenite. They also found that the Nb micro-alloyed steel had higher flow stresses and peak strains than the Nb-free steel, which could be attributed to the solute drag effect of Nb. A solute drag effect of Nb results in slower softening kinetics of Nb containing steel. To further refine the research on softening kinetics, an efficient optimization design based on TRIP steels for compression tests was carried out by Kuziak et al. [9]. They found that the dynamic recovery of the transformation-induced plasticity (TRIP) steels is a main thermally activated process occurring during hot compression tests. Evgueni et al. [10] investigated the hot deformation softening mechanisms of Dynamic Recrystallization Controlled Rolling, and found it was imperative to maintain the delicate balance between rolling temperature, speed, acceleration, and pass reduction, tailored to the specific steel. The designs of third generation AHSSs consist of complicated alloys, which are significant to the role of each manufacturing stage, therefore studying the Controlled Rolling and Controlled Cooling Process of AHSSs is of scientific importance, since the microstructure evolution is substantially different due to hot deformation behavior. Aydin et al. [7] investigated the hot deformation characteristics of AHSS plates with various Mn contents using hot rolled tests. They observed that finding accurate hot rolling parameters is a critical stage, responsible for producing a microstructure that is optimal for achieving the final properties of the sheet products. Zhang et al. [11] studied the hot strip rolling of a 0.16%C–0.8%Mn steel, and observed that the experimental steel was characterized by high resistance to hot deformation during hot strip rolling. 

## 2. Materials and Methods

In this study, a new third generation AHSS, namely Fe–5Mn–3Al–0.1C steel, was designed with the aid of JMatPro software, and hot compression tests were conducted using a Gleeble-3800 thermo-mechanical simulator. Based on the experimental data, the critical characteristic parameters were calculated, 3D processing maps were established, and relatively ideal hot processing parameters were determined. In addition, the microstructure evolution and DRX nucleation mechanisms were investigated using a range of microscopy techniques.

### 2.1. Sample Preparation

The chemical composition (% wt.) of the newly designed Fe–5Mn–3Al–0.1C high-strength lightweight steel is shown in Table 1. The experimental steel was prepared using vacuum induction melting, and was hot forged into a block. To obtain a homogeneous duplex structure consisting of austenite and δ-F, the block was homogenized at 1200 °C for 12 h, and then was furnace-cooled to room temperature. The microstructure of the experimental steel at room temperature consisted of δ-ferrite, austenite, and martensite, as shown in Figure 1, which suggests that martensite is formed through austenite–martensite phase transformation, however, δ-ferrite remains unchanged in the cooling process.

### 2.2. Hot Deformation

Cylindrical specimens of 8 mm in diameter and 12 mm in height were cut from the homogenized block. The isothermal uniaxial compression tests were conducted using a Gleeble-3800 thermo-mechanical simulator at different temperatures ranging from 900 to 1150 °C at intervals of 50 °C, and constant strain rates of 0.1, 1, 2.5, 5, 10, and 20 s^−1^. Prior to compression, the specimens were heated to a 1200 °C homogenizing treatment with a heating rate of 10 °C/s, held for 3 min, and then cooled to the compression temperatures at the rate of 10 °C/s. To achieve a homogeneous temperature distribution, the specimens were soaked for 30 s at the compression temperatures, and then compressed to 0.7 true strain. Finally, the deformed specimens were water quenched to room temperature. Figure 2 shows the schematic illustrating the hot deformation tests and the hot deformation process of the Fe–5Mn–3Al–0.1C steel. 

### 2.3. Microstructure Characterization

The specimens for OM (optical microscope) observation were sectioned parallel to the compression axis and were etched in a 25% sodium bisulfite solution for 2 min. The specimens for scanning electron microscope (SEM) analysis and electron backscattered diffraction (EBSD) analysis were also cut parallel to the compression axis, and they were electro-polished in a solution of 90% acetic acid and 10% chloric acid at 20 kV with a step size of 0.2 μm. Thin foils with a diameter of 3 mm for transmission electron microscopy (TEM) studies were mechanically thinned to 50 μm thickness from both sides and finally thinned using a twin-jet electropolished (StruersTenuPol-5) in a solution of 95% alcohol acid and 5% chloric acid. They were examined by an FEI Tecnai G2 F20 TEM operated at 200 kV.

The micro-hardness of the phases was tested on a Vickers Hardness Tester six times, and an average value was obtained.

## 3. Results and Discussion

### 3.1. Simulation in JMatPro

The phase transformation of the experimental steel was analyzed using JMatPro software, and then the equilibrium phase diagram was obtained, as shown in Figure 3. It can be clearly seen from Figure 3 that the ferrite, which is usually called high temperature δ-ferrite (δ-F), is formed in the liquid phase at about 1510 °C. When the temperature is lowered to 1439 °C, the liquid phase has completely transformed to δ-F. Then, the δ-F starts transforming to austenite at 1428 °C. The amount of austenite increases with the reduction of δ-F and reaches the maximum at 1103 °C. It is worth noting that δ-F does not completely transform to austenite, which can be ascribed to the fact that aluminum (Al) is a strong ferrite former and the addition of Al can stabilize the δ-F [3]. Subsequently, the austenite starts transforming to α-ferrite and then to martensite, until the temperature drops to 600 °C. From Figure 1, it is clear that the observed microstructure agrees quite well with the calculative one.

### 3.2. Physical Modeling on Gleeble

#### 3.2.1. Flow Behavior

It can be clearly seen from Figure 4 that steel is sensitive to deformation parameters. The most notable characteristic of steel is that multiple peak behaviors were seen in the resultant flow curves. This is called multiple transient steady-state (MTSS) behavior [12]. There exists a competition between dynamic recovery (DRV) or DRX in the hot deformation, and this fact is attributed to the occurrence of several independent cycles of DRX [13]. The curves could be segmented into three parts; for instance, the curve in Figure 4a with the deformation conditions of 900 °C and 0.1 s^−1^. There is deformation competition between δ-ferrite and austenite during hot deformation. The harder austenite provides greater plastic deformation resistance than the softer δ-ferrite. At stage I, the work-hardening of δ-ferrite, which is softer than austenite, is dominant due to the increase of dislocation density. This causes the rapid increase of flow stress at the beginning of deformation until it reaches the first peak value of σ_p1_. At stage II, the work-hardening (WH) of austenite becomes dominant until the dynamic balance between WH, and dynamic softening occurs [14,15]. In this stage, the flow stress gradually increases to the second peak value of σ_p2_. At stage III, the flow stress gradually falls or tends to maintain a dynamic balance, which is a typical DRX characteristic.

The micro-hardness was analyzed in order to illustrate the above phenomenon in detail. The Vickers hardness test results are shown in Figure 5. The micro-hardness of each phase is presented in Table 2. Figure 5a is the original structure of a forged Fe–5Mn–3Al–0.1C steel blank; the original structure consists of δ-ferrite and island-like austenite. Figure 5b is the microstructure of the sample deformed at the temperature of 950 °C and the strain rate of 0.1 s^−1^. The microstructure is composed of δ-ferrite and martensite transforming from austenite during the quenching process. The δ-ferrite and martensite are elongated via the compress process, and both of them display a banded microstructure. The micro-hardness of δ-ferrite increases from 143 to 246 HV. The micro-hardness of the initial austenite is 195 HV, which is comparable to the hardness of the austenite phase (212 ± 10 HV) in research conducted by Cai et al. [16]. It is reasonable to infer that the plastic deformation occurs first in the softer δ-ferrite phase, and then in the harder austenite phase. DRX occurs when the strain energy and interface energy accumulate to the critical values. 

#### 3.2.2. Determination of DRX Critical Stress and Strain

It is well known that DRX occurs when the strain accumulates to some extent before the peak strain. In other words, there is a threshold strain which is called the critical strain [15,17]. Based on thermodynamic irreversibility theory [18] and previous research results [19,20], the ‘double differential method’ was summarized by Poliak and Jonas [21,22], according to which the following DRX critical stress equation is derived [23]:(1)$$\left\{\begin{array}{l} f(\theta ,\sigma )=\frac{\partial }{\partial \sigma }\left(\frac{\partial \theta }{\partial \sigma }\right)\\ \theta ={\left.\frac{\partial \sigma }{\partial \varepsilon }\right|}_{T,\dot{\varepsilon }} \end{array}\right.$$
where *θ* is the work hardening rate. DRX starts when f(θ,σ)=0. The Zener–Hollomon (*Z*) parameter Equation (2) is usually used to describe DRX behaviors, and it can reflect how difficult DRX occurs [24,25].
(2)$$Z=\dot{\varepsilon }\exp\left(\frac{Q}{RT}\right)=A{\left[\sinh(a\sigma )\right]}^{n}$$
where *R* is the gas constant (8.31 J∙mol^−1^∙K^−1^); *T* is the deformation temperature; and *Q* is the activation energy (kJ∙mol^−1^), which represents the difficulty level of an energy barrier to be surmounted in atomic diffusion. The value of *Q* is strongly affected by the alloy content, distribution of particles, and microstructure [26]. *Z* can be obtained by an Arrhenius constitutive model, and some previous work [27] has concentrated on the detailed solution procedure of *Z*. The critical stress model of DRX is expressed as follows:(3)$$\left\{\begin{array}{l} \sigma_{c}=\alpha_{1}Z^{a_{2}}\\ \sigma_{p}=\alpha_{3}Z^{a_{4}} \end{array}\right.$$
where *σ*_c_ is the critical stress; *σ*_p_ is the peak stress; and *a*_1_, *a*_2_, *a*_3_, and *a*_4_ are constants. Regression analysis of Equation (3) results in the following equation:(4)$$\left\{\begin{array}{l} \ln\sigma_{c}=\ln a_{1}+a_{2}\ln Z\\ \ln\sigma_{p}=\ln a_{3}+a_{4}\ln Z \end{array}\right.$$

The *θ* vs. *σ* and *θ* vs. *ε* curves are obtained as shown in Figure 6. The critical points of DRX can be found at the inflection points of these curves.

According to the ‘double differential method’ criterion [15,24], when the condition $\frac{\partial (-\partial \theta /\partial \sigma )}{\partial \sigma }=0$ is met, DRX occurs. Therefore, the critical strain corresponds to the maximum value of the (d*θ*/d*ε*) − *ε* curve shown in Figure 7a. The critical stress corresponds to the minimum value of the (d*θ*/d*ε*) − *σ* curve shown in Figure 7b. The critical values of the stress and strain required for the initiation of DRX were calculated at various deformation parameters and are given in Figure 7c. It is clear from these data that the critical stress decreases with the increase in temperature, and increases with the increase in the strain rate. This microstructure, with complex dynamic deformation mechanisms, was studied by Wen et al. [24]. They found that the deformation temperature plays an essential role in the dislocation movement mechanism. Higher temperatures benefit the recombined active dislocations to dislocation cells and low angle grain boundaries (LAB) by the dislocation motion, and then DRX grains form by multilateralization and LAB transformation to high angle grain boundaries, respectively [28]. Furthermore, Liu et al. [25,29] found that there is ample time to prepare dislocation recombinations under the relatively low strain rate. Therefore, it is reasonable that the critical strain decreases with the increase of the deformation temperature.

Figure 8 shows the plots of ln*σ*_p_–1n*Z* and 1n*σ*_c_–1n*Z*. As can be seen, the values of *a*_2_ and *a*_4_ are the slopes of the plots of 1n*σ*_p_–1n*Z* and 1n*σ*_c_–1n*Z*. The values of ln*a*_1_ and ln*a*_3_ are the intercepts of the two plots, respectively.

Through the above analysis, the DRX critical strain and critical stress models can be obtained as follows:(5)$$\left\{\begin{array}{l} \begin{array}{l} \varepsilon_{c}=1.13Z^{0.00031}\\ \varepsilon_{p}=1.066Z^{0.00326} \end{array}\\ \begin{array}{l} \sigma_{c}=11.276Z^{0.0436}\\ \sigma_{p}=11.878Z^{0.00434} \end{array} \end{array}\right..$$

### 3.3. Processing Map and Microstructure Analysis

The processing map has been generalized to optimize the process parameters during hot deformation [4,30]. The thermal compression testing process can be considered as a power dissipation system; the total compression power (*p*) can be divided into two sections: *G* and *J*. [31,32]
(6)$$P=\sigma \dot{\varepsilon }=G+J=\int_{0}^{\dot{\varepsilon }}\sigma d\dot{\varepsilon }+\int_{0}^{\sigma }\dot{\varepsilon }d\sigma$$
where *G* is used to describe the energy consumed via the elastic-plastic deformation, and *J* represents the consumption of the microstructural changes. The efficiency of power dissipation (*η*) is given by:(7)$$\left\{\begin{array}{l} \eta (T,\dot{\varepsilon })=\frac{J}{J_{\max}}={\left.\frac{2m}{m+1}\right|}_{\dot{\varepsilon }}\\ m={\left(\frac{dJ}{dG}\right)}_{\varepsilon ,T}=\left(\frac{\partial \ln\sigma }{\partial \ln\dot{\varepsilon }}\right) \end{array}\right.$$
where *m* is strain rate sensitivity index. Based on the continuum instability criterion proposed by Prasad and Kumar [33], the flow instability criterion is given by:(8)$$\left\{\begin{array}{l} \eta (T,\dot{\varepsilon })=\frac{J}{J_{\max}}={\left.\frac{2m}{m+1}\right|}_{\dot{\varepsilon }}\\ m={\left(\frac{dJ}{dG}\right)}_{\varepsilon ,T}=\left(\frac{\partial \ln\sigma }{\partial \ln\dot{\varepsilon }}\right) \end{array}\right.$$

Based on the above theories, the 3D power dissipation maps and 3D instability maps of the newly designed Fe–5Mn–3Al–0.1C steel at various deformation parameters were established (Figure 9). As presented in Figure 9a, the contour color represents the percentage of power-dissipation efficiency, which is sensitive to strain. With increases in strain, the proportion of low-dissipation regions (the dark blue regions) gradually decreases. As presented in Figure 9b, the light blue region reflects stability, while the magenta region is instable. The instability region becomes smaller progressively with the increase in strain. Furthermore, 2D hot processing maps were established by superimposing the power dissipation and instability maps in Figure 10. In this figure, gray regions are the flow instability regions, which are usually called unsafe regions [34]. The power dissipation values (*η*) are denoted by contour numbers. Comparing the four processing maps with different strains, one can see that η in the stability regions is higher than in the instability regions, which indicates that sufficient energy consumption is provided for DRX within these regions. Figure 10d is a four-part figure that illustrates the microstructure evolution in different regions at a strain of 0.7. It can be clearly seen from Figure 10d that there are mainly two safe regions and two unsafe regions.

#### 3.3.1. Region A

Region A is an instable area occurring in the temperature range of about 900 to 930 °C and the strain rate range of approximately 0.16 to 20 s^−1^. Tiny cracks present along the δ-ferrite grain boundaries, which can be clearly observed in Figure 11. As the strain rate increases, as observed in Figure 11a,b, the micro-cracks extend and longitudinal cracks are formed along the ferrite phase boundaries. The cracks extend to the surface when the strain rate reaches 20 s^−1^, which may lead to the fracture of specimens, as observed in Figure 11c. Note that the cracks are easily generated from the carbides located at the grain boundaries, which will lead to the occurrence of quasi-cleavage or cleavage fracture modes [34]. Research suggests that κ-carbides formed by Fe, Mn, C, and Al can be found in medium manganese automobile steel, which is not good for the stability of the microstructure [35,36].

#### 3.3.2. Region B

Region B is a flow instability area occurring in the temperature range of approximately 1075 to 1200 °C and the strain rate range of 0.22 to 20 s^−1^. Figure 12a and b show the EBSD band contrast maps of the samples deformed at the strain rate of 20 s^−1^ and the temperatures of 1000 and 1200 °C, respectively. DRX grains with a size of approximate 10 μm are formed in most of the observed areas, as shown in Figure 12a. By contrast, inhomogeneous DRX grains are observed in Figure 12b. This can be attributed to the fact that a higher temperature provides a higher driving force for the mobility of grain boundaries, which promotes the growth of DRX grains [20]. In order to determine whether the DRX of δ-ferrite has completed, the EBSD boundary map and misorientation distribution map of the specimen with inhomogeneous grains are plotted (Figure 12c,d). As can be clearly seen in Figure 12c,d, the low angle grain boundaries (lower than 5°) account for about 58.7%, and they distribute mainly in martensite due to incomplete dynamic recrystallization. However, the grain boundary angles of δ-ferrite are mainly larger than 5°, and such a mixed microstructure is usually considered a defect structure.

#### 3.3.3. Region C

Region C is a flow stability area occurring in the temperature range of approximately 950 to 1150 °C and the strain rate range of 0.1 to 0.2 s^−1^. Figure 13 shows the microstructures of the specimens deformed at temperatures of 950, 1050, and 1150 °C, and a constant strain rate of 0.1 s^−1^. In this region, the dissipation efficiency was much higher, amounting to η ≤ 38%. However, as a result of a too low a strain rate, DRX occurs obviously inside δ-ferrite, in austenite, DRX is rarely observed since the critical stress and strain for the initiation of DRX of austenite are higher than those of δ-ferrite. The initiation of dynamic recovery (DRV) of austenite. It can also be observed in Figure 13 that the volume fraction of dynamic recrystallized δ-ferrite increases with the increase in temperature. This can be ascribed to the fact that a higher temperature will provide a higher nucleation driving force and facilitate the DRX of δ-ferrite.

#### 3.3.4. Region D

Region D is a flow stability area occurring in the temperature range of approximately 950 to 1050 °C and the strain rate range of 0.54 to 20 s^−1^. In this region, a higher level of power dissipation efficiency (19.9–28.3%) exists. Figure 14a and b shows the microstructures and misorientation angle distributions of the specimens deformed at 1000 °C and strain rates of 10 and 20 s^−1^. The misorientation angle distributions display a typical bimodal type. The number fractions of 2–10° misorientation angles are 61.6% and 48.4% for the strain rates of 10 and 20 s^−1^, respectively. The number fractions of 50–60° misorientation angles are 15.5% and 17.1% for the strain rates of 10 and 20 s^−1^, respectively. It can also be observed in Figure 14a and b that a larger number of DRX grains are formed at grain boundaries when the stain rate reaches 20 s^−1^. To quantitatively analyze the volume fraction of DRX grains, the DRX degree were determined by the Channel-5 attached to EBSD Analysis software. Using the Recrystallized Fraction Feature module in Channel-5 to obtain the volume fraction of dynamic recrystallization. 3D DRX maps were plotted using the convert HKL-Salsa, as shown in Figure 14c and d. The ratio of DRX grains increases from 19.9% to 28.3% when the strain rate increases from 10 s^−1^ to 20 s^−1^, which indicates that it is easier for DRX to occur at the strain rate of 20 s^−1^. This can be attributed to the fact that a higher strain rate will result in the increase of dislocation density, which contributes to the increase of the driving force for recrystallization.

### 3.4. DRX Mechanism of Stable Regions

The nucleation mechanism of DRX is particularly important for precise microstructure control of the newly designed Fe–5Mn–3Al–0.1C steel. To study the nucleation mechanism, EBSD analysis and TEM analysis were carried out, and the results are shown in Figure 15. The local misorientation (point-to-point) and the cumulative misorientation (point-to-origin) are obtained along the P1and P2 lines marked in Figure 15a, and it was analyzed in detail in Figure 15b, It can be clearly observed in Figure 15b that the orientation angle increases gradually from 7° to 21° along the direction indicated by the white arrow P1, and the misorientation angle of the adjacent grains is between 2° and 7°, which means that the homogenous sub-boundaries are formed in the deformed δ-ferrite grains. Figure 15c also proves the formation of sub-boundaries.

Figure 15d shows the TEM image of δ-ferrite. Dislocation accumulation and tangle can be clearly observed. Dislocation cells are formed due to dislocation tangle, and then evolve into sub-grains by continuously absorbing the surrounding dislocations. Successively, the sub-grains continue to absorb dislocations and gradually increase the orientation differences with the adjacent sub-grains, and then the low-angle boundaries transform into large-angle grain boundaries. Finally, new DRX grains are formed [37]. Therefore, the nucleation mechanism of δ-ferrite is a typical CDRX mechanism.

From Figure 15e, it is clear that the orientation angle increases gradually from 15° to 60° along the direction indicated by the white arrow P2, and the misorientation angle of the adjacent grains is between 2° and 60°, which indicates that the distribution of the misorientation angle is random. A lot of large-angle grain boundaries are present in the adjacent δ-ferrite grains, as shown in Figure 15f. It is well known that the vicinities of grain boundaries are the preferential sites of DRX. Therefore, these grains are typical DRX grains, which result in the random distribution of the grain boundary angle. This is consistent with the above analysis.

Figure 15g shows the TEM image of martensite. It can be observed in this figure that the small nuclei of DRX grains are formed at the bulging grain boundaries. The adjacent grains exhibit inhomogeneity of dislocation density, which is very important to the stability of grain boundaries during hot deformation. This internal difference will drive the grain boundaries to move from high-density areas to low-density ones, resulting in the bulge of the grain boundaries into low-density areas [38]. When meeting certain conditions, DRX grains will form and grow, as seen in Figure 15h. The grain boundary bulging can also lead to the formation of necklace morphology, as shown in Figure 15i. Therefore, the nucleation mechanism of martensite is grain boundary bulging [39,40].

## 4. Conclusions

The hot deformation behavior and the underlying DRX mechanisms in the new designed Fe–5Mn–3Al–0.1C High-Strength Reduced-Density Steel for automobiles were analyzed. The micro-structure of the starting material, before thermomechanical treatment, was composed of δ-ferrite and austenite. Hot compression tests were designed in the temperature range of 900 to 1150 °C, at constant strain rates of 0.1, 1, 2.5, 5, 10, and 20 s^−1^. The following conclusions can be drawn:The DRX critical strain (stress) predicting model, considering the Zener–Hollomon (Z) parameter and peak strain (stress), was established.
(9)$$\left\{\begin{array}{l} \begin{array}{l} \varepsilon_{c}=1.13Z^{0.00031}\\ \varepsilon_{p}=1.066Z^{0.00326} \end{array}\\ \begin{array}{l} \sigma_{c}=11.276Z^{0.0436}\\ \sigma_{p}=11.878Z^{0.00434} \end{array} \end{array}\right.$$

It was found to be very significant for the rolling schedule design.

2.The processing maps of the newly designed Fe–5Mn–3Al–0.1C with different strains were constructed. With increasing strain, the low-dissipation region gradually becomes smaller, which means the processing region becomes larger. This occurs under the temperatures of 1000–1050 °C and the strain rate of 10–20 s^−1^.3.The original microstructure of the steel consists of δ-ferrite and austenite. There is formation of DRX new grains within δ-ferrite grains and austenite grains during the hot compression test. The steel possesses unfavorable toughness at the deformation temperature of 900 °C, which is mainly due to the presence of coarse κ-carbides along grain boundaries, as well as the lower strengthening effect of grain boundaries.4.The local misorientation (point-to-point) and the cumulative misorientation (point-to-origin) of EBSD and TEM image were analyzed in detail, there are two different DRX nucleation mechanisms that operate during the compression deformation process. DDRX is the primary nucleation mechanism of austenite, and DRX nuclei form due to the local bulging out of the original grain boundaries. The nucleation mechanism of CDRX was found in δ-ferrite, which is due to dislocation tangling.

## Figures and Tables

**Figure 1 materials-14-02478-f001:**
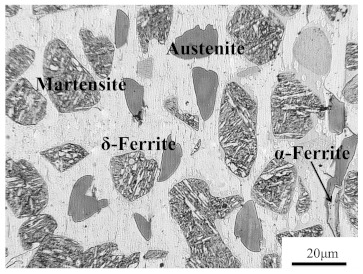
Microstructure of experimental steel.

**Figure 2 materials-14-02478-f002:**
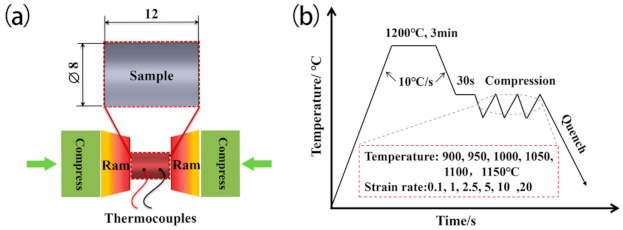
(**a**) Schematic plot of the hot deformation tests and (**b**) schematic plot of the hot deformation process tests.

**Figure 3 materials-14-02478-f003:**
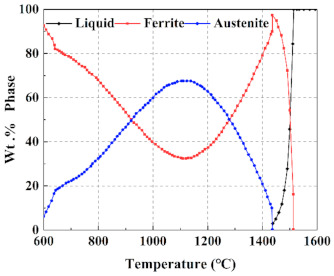
Equilibrium phase diagram of experimental steel.

**Figure 4 materials-14-02478-f004:**
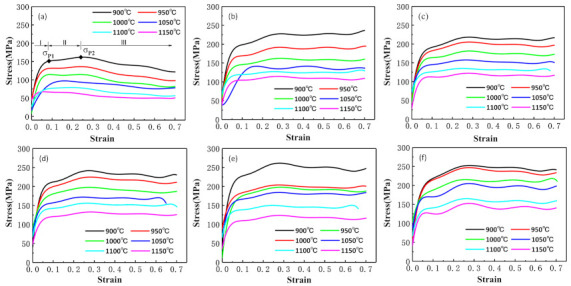
True stress–strain curves at various deformation parameters: (**a**) 0.1 s^−1^, (**b**) 1 s^−1^, (**c**) 2.5 s^−1^, (**d**) 5 s^−1^, (**e**) 10 s^−1^, and (**f**) 20 s^−1^.

**Figure 5 materials-14-02478-f005:**
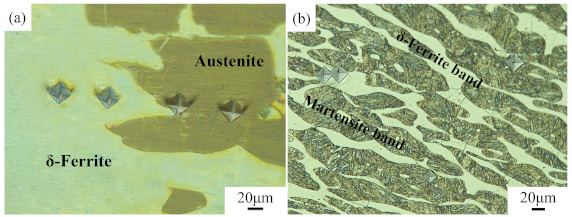
(**a**) Vickers hardness test of the initial microstructures and (**b**) Vickers hardness test under deformation conditions of 950 °C and 0.1 s^−1^.

**Figure 6 materials-14-02478-f006:**
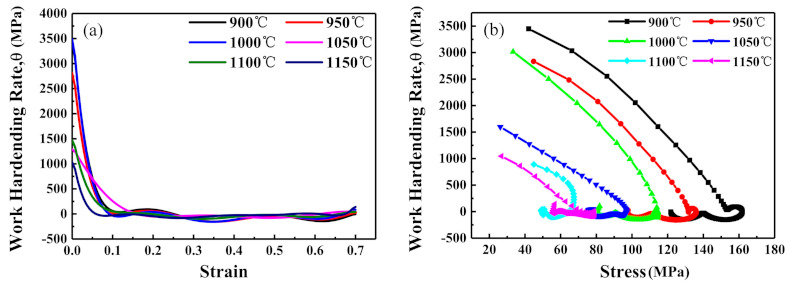
(**a**) *θ*−Strain and (**b**) *θ*−*σ* curves under different temperatures and a strain rate of 0.1 s^−1^.

**Figure 7 materials-14-02478-f007:**
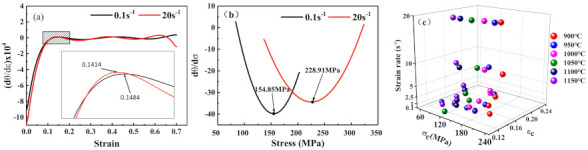
Variation of (**a**) d(*θ*)/d(*σ*)—strain, (**b**) d(*θ*)/d(*σ*)—stress at 900 °C, and (**c**) the plot of *σ*_c_ and *ε*_c_ under different conditions.

**Figure 8 materials-14-02478-f008:**
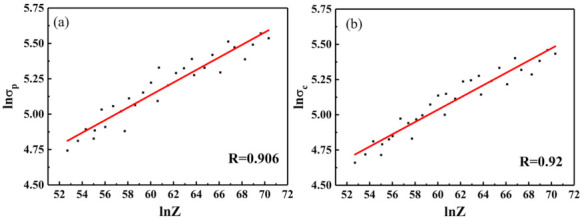
Plot of (**a**) 1n*σ*_p_-1n*Z* and (**b**) 1n*σ*_c_-1n*Z*.

**Figure 9 materials-14-02478-f009:**
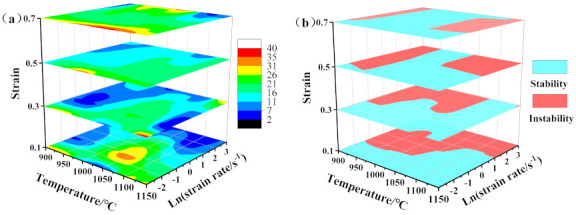
3D power dissipation maps (**a**) and instability map (**b**) of the newly designed Fe–5Mn–3Al–0.1C.

**Figure 10 materials-14-02478-f010:**
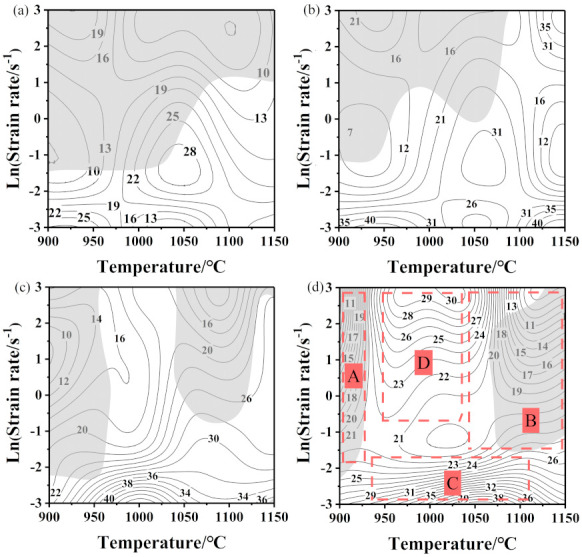
Processing maps under strains of (**a**) 0.1 s^−1^, (**b**) 0.3 s^−1^, (**c**) 0.5 s^−1^, and (**d**) 0.7 s^−1^.

**Figure 11 materials-14-02478-f011:**
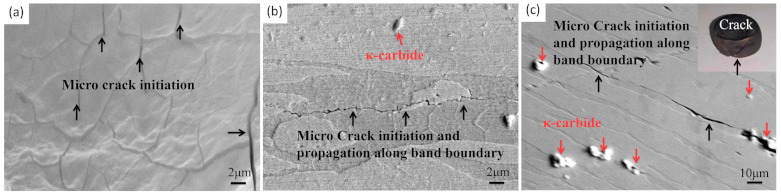
SEM images under conditions of (**a**) 900 °C, 1 s^−1^, (**b**) 900 °C, 2.5 s^−1^, and (**c**) 900 °C, 20 s^−1^.

**Figure 12 materials-14-02478-f012:**
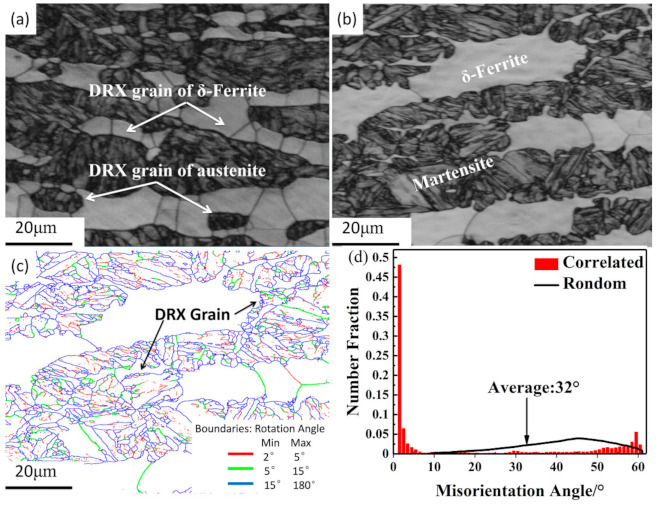
EBSD band contrast maps under the conditions of (**a**) 1000 °C and 20 s^−1^, (**b**) 1200 °C and 20 s^−1^, (**c**) the grain–subgrain boundary map, and (**d**) misorientation distribution map of 1200 °C and 20 s^−1^.

**Figure 13 materials-14-02478-f013:**
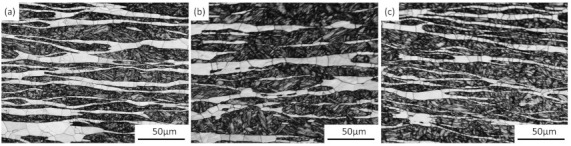
EBSD band contrast maps under the conditions of (**a**) 950 °C, (**b**) 1050 °C, and (**c**) 1150 °C with a strain rate of 0.1 s^−1^.

**Figure 14 materials-14-02478-f014:**
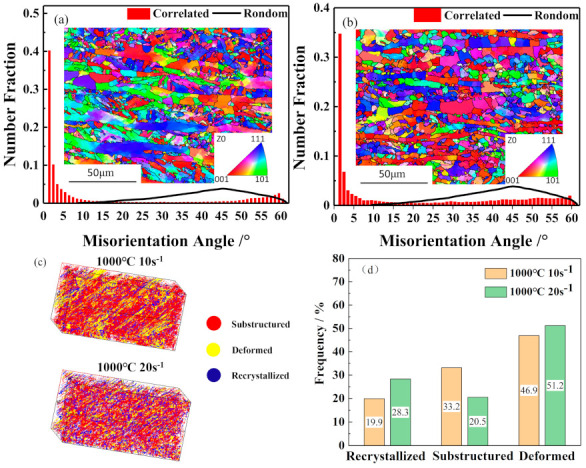
IPF map and misorientation distribution under conditions of (**a**) 1000 °C and 10 s^−1^, (**b**) 1000 °C and 20 s^−1^ and their 3D DRX maps, (**c**) recrystallization ratio, and (**d**) the volume fraction of dynamic recrystallization.

**Figure 15 materials-14-02478-f015:**
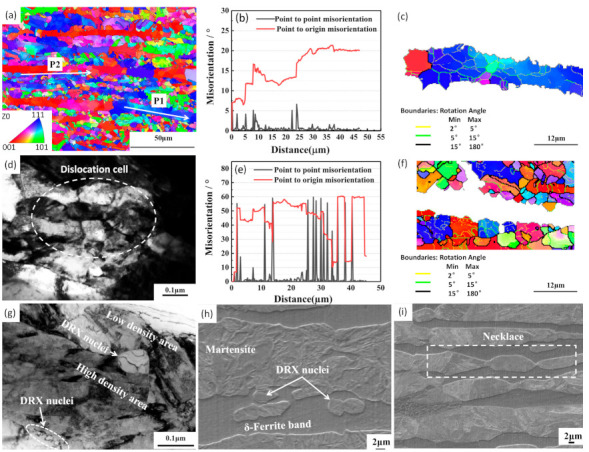
DRX behavior under deformation conditions of 1000 °C and 20 s^−1^. (**a**) IPF map, (**b**) grain boundary misorientation angle in δ-ferrite, (**c**) grain and sub grain boundary map of δ-ferrite, (**d**) formation of dislocation cell, (**e**) grain boundary misorientation angle of austenite, (**f**) grain and sub grain boundary map of austenite, (**g**) formation of DRX nuclei, (**h**) DRX grains nucleate adjacent serrated GBs, and (**i**) grain necklaces along original grain boundaries.

**Table 1 materials-14-02478-t001:** The chemical composition of the Fe–5Mn–3Al–0.1C steel (% wt.).

Element	C	Mn	Al	Fe
Content (% wt.)	0.13	5.50	3.10	Bal.

**Table 2 materials-14-02478-t002:** Microhardness of the constituent phases.

Phases	Vickers Hardness (HV)
Initial δ-ferrite	143 ± 5
Initial austenite	195 ± 8
δ-ferrite band	246 ± 6
Martensite	372 ± 10

## Data Availability

All data generated or analyzed during this study are included in this article.

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
