# Peer review of "Hot Deformation Behavior and Microstructure Evolution of Fe–5Mn–3Al–0.1C High-Strength Lightweight Steel for Automobiles"

_materials, 2021, doi:10.3390/ma14102478_

Round 1

Reviewer 1 Report

Dear authors,

This is a very good paper with new and interesting results. My impression is that this work is a very solid piece of science and an important one. The paper concerns the optimization of (hot) forming parameters based on the efficiency of power dissipation, called as processing maps technique. Obtaining a material with the best possible mechanical properties through the hot working is an important task.

The article is related to the setting of the optimal parameters of processing the base material and the modified modern energy analysis by the process of plastic deformation and its attendant structural effects. This analysis is carried out in a very professional and extensive manner. Its results are verified on the basis of experimental studies of the obtained structures, and the discussion combines both these aspects.

However, in an article I noticed some shortcomings, which does not affect my positive opinion. I have the following reservations. The authors in their analyzes go beyond the area of the analyzed cases defined by the parameters of hot working. The scattering of data about the experiment and test conditions in the text makes it difficult to control this fact, and at the same time creates the impression of a compositional disorder. Both in the description of the experiment methodology and in Fig. 2b, the indicated parameters do not apply to the temperature of 1200 °C and deformations with a speed greater than 0.1 s-1. Meanwhile, in Fig. 9 and above all 10, the presented processing maps go beyond the areas of the experiment. This is not allowed, as they are only presumptions which do not have to be true.

Less important comments and suggestions in the attached file. 

Reviewer 2 Report

The study is devoted to a topical topic, modern methods and research equipment were used in the work. However, the presentation of research results does not allow the reader to understand the progress of the work and is difficult to perceive. The work is poorly structured. The article needs to be completely recompiled. The article mixes an overview part and a research part. Now the article contains two sections of the same name 2. and 3. Materials and methods. 

I suggest the authors divide the article into the following sections:

  1. Introduction.
  2. Materials and methods.
  3. Results and discussion: 3.1. Simulation in JMatPro. 3.2 Physical modeling on Gleeble. 3.3. Microstructure analysis.
  4. Conclusions.

The conclusions of the work should also be specified.

In addition, table 1 shows the chemical composition of the investigated steel. Why does it start with Al in the table, and not with C as is usually accepted? It is better to give the sequence of elements in the accepted order: C, Mn, Al ... Is there no silicon (Si) in the investigated steel at all?      

Round 2

Reviewer 1 Report

Dear authors,

The comments I have submitted are only a form of discussion of the matter. They mainly result from a misunderstanding of the presented content. So you are for a sign that is what you wrote some reason it is incomprehensible to the reader. But it may also be the result of mistakes in your interpretation of the admittedly difficult issues. That is why the review of the person standing on the sidelines is so important for the quality of the description of the phenomena presented. I am glad that my comments found your approval. And the wide range of corrections you have introduced positively affects the quality of the article.

I just want to add that the phrase "reduced-density steel" is only a statement of the actual state and does not have to be appropriate for the vocabulary used here. Consider this point and make the right decision.

I also believe that it would be appropriate to mention in the abstract that it is about weight composition - add the acronym: (wt%).

Reviewer 2 Report

The article after the elimination of comments has become much better. In this form, the article can be accepted for publication.

Author Response

Dear Reviewers:

We would like to express our sincere appreciation for your careful reading and invaluable comments that have helped improve this paper substantially.

Kind regards,
Mr. Wang

This manuscript is a resubmission of an earlier submission. The following is a list of the peer review reports and author responses from that submission.

Round 1

Reviewer 1 Report

Detailed structural studies of the phase composition transformations in new AHSS steel have been carried out in a wide range of temperatures and strain rates. All obtained results are scientifically substantiated and novel. The theoretical processing and interpretation of the results is reliable. The article needs to be made minor corrections.

  1. Figure 1 is a modification of a previously published figure in I. von Hagen, in: Proceedings of the Conference on Steels in Cars and Trucks,June 05–10 2005, Stahleisen, Wiesbaden, Germany. A link to this publication should be included in the caption to Figure 1.
  2. The Fe-5Mn-3Al-0.1C steel studied in the article is very close to that developed earlier in https://doi.org/10.1016/j.pmatsci.2018.01.006. In this steel, the authors discovered delta ferrite. It is necessary to submit the results of tensile tests conducted on hot rolled and annealed plates in order to confirm that this steel belongs to the 3rd generation AHSS.
  3. In conclusion does not need to list what was investigated. The new obtained scientific results should be presented in a concise form. The following phrases should be removed from the text of the Conclusion:

- The yield-point-elongation-like effect was observed;

- The initiation DRX critical strain (stress) data with different hot deformation conditions was obtained.

- DRX behaviors and effects is discussed;

- The DRX mechanism of the relatively ideal processing region was studied;

These phrases are typical for Abstract but not for Conclusions.

Reviewer 2 Report

Manuscript entitled “Hot deformation behavior and microstructure 2 evolution of Fe-5Mn-3Al-0.1C high-strength low-3 density steel for automobile” presented the new hight-strength low-density steel for automobile. The manuscript is very interesting and desirable with application potential. In order to publish the manuscript, major revision is not required to improve the quality of the publication. My comments are written below:

Figure 2 b: needs bigger fonts for clarity.

Figure 4 b, c d, e, f: change Mpa → MPa.

Figure 5 c: needs bigger fonts for clarity.

Reviewer 3 Report

The topic of the paper corresponds to the "Materials" Journal.
The research idea is of real scientific and practical interest. The paper content is well structured and clearly presented. The presented experiments of the paper are reliable.
The authors should better and more clearly emphasise the novelty of their research and how the results will improve the scientific knowledge in the field approached.
The abstract should be rewritten in order to clearly and briefly describes the topic of the study, the problem, the investigation methodology, the most important findings, implications, as well as few concise and precious conclusions of the research.
The chapter of conclusions should be improved to better argue the use of Fe-5Mn-3Al-0.1C steel in automobile construction
